# Analysis of Single-Leg Hopping in Long-Term Treated Patients with Neurological Wilson’s Disease: A Controlled Pilot Study

**DOI:** 10.3390/medicina58020249

**Published:** 2022-02-07

**Authors:** Harald Hefter, Sara Samadzadeh, Dietmar Rosenthal, Osman Tezayak

**Affiliations:** 1Department of Neurology, University of Düsseldorf, Moorenstrasse 5, D-40225 Düsseldorf, Germany; sara.samadzadeh@yahoo.com (S.S.); dietmar.rosenthal@med.uni-duesseldorf.de (D.R.); tezayak@hispeed.ch (O.T.); 2Department of Psychiatry, Psychiatriezentrum Kreuzlingen, Nationalstrasse 19, CH-8280 Kreuzlingen, Switzerland

**Keywords:** Wilson’s disease, single-leg hopping, limb/trunk coordination, ground reaction forces, optimization of therapy

## Abstract

*Background and Objectives*: In long-term treated patients with neurological Wilson’s disease, the ability to perform single-leg hopping was analyzed to quantify motor deficits. *Materials and Methods*: Twenty-nine long-term treated Wilson patients had to stand on one leg for at least 3 s and then perform at least five consecutive hops on this leg. Ground reaction forces and temporal patterns of hopping were recorded using an Infotronic^®^ walking system, which consists of soft tissue shoes with a solid, but flexible plate containing eight force transducers allowing measurement of ground reaction forces (GRF) and temporal patterns of foot ground contact. Parameters of hopping were correlated with clinical scores and parameters of copper metabolism and liver enzymes. Patients’ hopping data were compared with those of an age- and sex-matched control group. *Results*: Five severely affected Wilson patients were unable to hop. Time to the peak was significantly (*p* < 0.03) shorter in the remaining 24 patients compared to controls, but there was no difference in hopping frequency, the amplitude of ground forces and duration of foot contact. Twelve patients produced a second, sharp, initial “impact” force peak during ground contact in addition to the usual “active” force peak. Variability of the amplitude of the “active” peak was significantly inversely correlated with urinary copper elimination. *Conclusions*: The majority of long-term treated patients with neurological Wilson’s disease was able to perform single-leg hopping. The presence of a sharp initial “impact” peak in the GRF-curves of hopping may indicate a mild deficit of limb/trunk coordination and subclinical cerebellar impairment.

## 1. Introduction

Wilson’s disease (WD) is a rare, recessively inherited deficit of copper transportation [1,2,3] with increased serum levels of free copper and progressive copper intoxication of multiple organs [4]. In the central nervous system (CNS), the function of the basal ganglia, the cerebellum as well as the brain stem nuclei are predominantly affected [5]. Therefore, a broad spectrum of neurological, especially motor, symptoms result, among which (parkinsonian) gait, tremor, dysarthria, dystonia and chorea are observed quite frequently [6]. Frequencies of occurrence of initial neurological symptoms are reported in several large case series [7,8,9,10,11]. However, these reported frequencies vary widely [6]. The reported frequency for abnormal gait varies between being absent, less than 10% [12] or up to 75% [7].

The initial symptoms after neurological manifestation of WD respond differently to WD-specific therapy [6]. Response to therapy heavily depends on the patient’s adherence to therapy [13,14]. Tremor appears to be a frequent initial symptom that disappears during continuous copper elimination therapy but reappears when the patient’s compliance declines [6,13,14]. This is probably also the case with gait abnormalities. However, subtle gait abnormalities are more difficult to detect and analyze than tremors. In free-walking, WD patients’ quantitative analysis of foot ground reaction forces (GRF) revealed a mild reduction of gait speed [15], a reduced modulation of ground reaction force curves and a mild deficit of pushing with the forefoot at the terminal stance phase [16]. This subtle forefoot and big toe motor dysfunction was undetectable by simple observation during a clinical examination. Therefore, the present study focuses on single-leg hopping as a special motor task that tests forefoot and big toe function more intensively.

Hopping requires skillful organization of the spatiotemporal relationship between body segments [17] and is thought to be a much more complicated task to learn and perform than walking and running [18]. By about 3 years of age, lower limb joint motion during walking and running develop adult-like consistency [19]. However, at age seven, only about 60% of healthy children are able to hop on one leg [20]. It may take a further 5 years until the hopping motor skill is fully developed [18]. It was previously demonstrated that single-leg hopping imposes higher demands on motor competence and can discriminate between well and less coordinated children [18]. Single-leg hopping is used in various, difficult to analyze diseases such as children with hearing impairments [21], children with autism [22] and children with Down syndrome [23] to detect subtle motor coordination and balance problems. For the same reason, we analyze single-leg hopping in long-term treated patients with Wilson’s disease in the present study.

In healthy subjects and children with hearing impairments, single-leg hopping shows excellent reliability [21,24]. In clinical practice, standing and single-leg hopping are also used as a screening method to look for balance and motor control deficits and weakness of leg muscles [18,24]. The reason probably is that in a (good) first approximation, single-leg hopping can be described by a one-body mass-spring system with a natural resonance frequency depending only on the stiffness and the mass of this system [25,26,27]. Interestingly, man prefers the same frequency of single-leg hopping used by a kangaroo of similar size and the same frequency that an antelope of similar body mass prefers as stride frequency during galloping [26].

In patients and healthy subjects, hopping and especially the frequency of hopping is highly reproducible [21,24] as long as the different segments of the body can be coordinated as one unit and stiffness is not increased. This implies that disturbed hopping indicates an impairment of the ability to coordinate different segments of the body and/or to regulate muscle stiffness properly.

Therefore, the present pilot study on single-leg hopping in long-term treated WD was performed to analyze to what extent patients recover during long-term WD-specific treatment and whether a mild deficit of limb and/or trunk coordination persists.

## 2. Materials and Methods

### 2.1. Patients and Controls

Thirty WD patients who were treated as out-patients in the Department of Neurology of the University of Düsseldorf were consecutively recruited and gave their informed consent to participate in the present study on single-leg hopping. These 30 WD patients were able to walk a distance of at least 40 m without any aids and did not have a medical history for a walking deficit before the age of five. The study was performed according to the Declaration of Helsinki and the guidelines for good clinical practice (GCP) and was approved by the local ethics committee of the University of Düsseldorf (study number: 5171).

After all 30 WD patients were completely examined, an age- and sex-matched control group was recruited. Relatives of patients were excluded from the control group because of the possibility to be a gene carrier.

### 2.2. Neurological Examination of Patients and Clinical Scores

WD patients and controls underwent a detailed neurological examination. For each of the seven motor symptoms, dystonia, dysarthria, bradykinesia, tremor, gait disturbance, oculomotor deficits, ataxia of extremities and each of the three non-motor symptoms reflex abnormalities, sensory symptoms, neuropsychological and psychiatric symptoms, a score value was assigned. The score value was 0 when the symptom was absent, was 1 as long as the symptom was mild, 2 when it was moderate and 3 when it was severe. The 7 motor scores were added to yield a motor score (MotS: 0–21), the three non-motor items yielded a non-motor score (N-MotS: 0–9). MotS plus N-MotS yielded the total score (TSC: 0–30). This score was used in previous studies on WD [14,16,28,29]. A similar score is used by the Italian study group on liver transplantation in WD [30]. TSC was zero in all controls.

As in two previous studies [14,15], patients were split up into 3 different subgroups according to severity: mildly affected patients (MIL-group; TSC: 0–2; *n* = 10), moderately affected patients (MOD-group; TSC: 3–6; *n* = 11) and severely affected patients (SEV-group; TSC > 6; *n* = 9). The limits of the severity ranges were chosen to achieve a fairly equal subgroup size.

### 2.3. Laboratory Findings

After the clinical examination, blood samples were taken, and the 24 h-urine collected under medication was analyzed for patients’ routine monitoring of copper elimination therapy in WD. For correlation with hopping parameters, the following 10 parameters were chosen: liver enzymes (GOT, GPT, GGT), parameters of copper metabolism (serum level of copper, serum level of ceruloplasmin, copper concentration in the 24 h-urine), international normalized ratio (INR), thromboplastin time (PPT), platelet counts and creatinine.

### 2.4. Measurement of Hopping by Means of the Infotronic^®^ Gait Analysis System

The Infotronic^®^ (CDG^®^) gait analysis system (ref. [31]; NL-7650 AB Tubbergen, The Netherlands), which consists of soft tissue shoes with a solid, but flexible plate containing 8 force transducers, allows quantitative measurement of ground reaction forces (GRF-curves) and temporal patterns of foot ground contact. These shoes are strapped over the street shoes of the patients. Thin cables connect the CDG^®^-shoes with a light microprocessor which was tightly attached to a belt strapped around the hips of the patients. Patients and controls were allowed to perform up to 3 short test hops to become familiar with the set-up. An image of the shoes and a subject wearing the shoes are presented in [16]. We used two pairs of shoes of different sizes because of the different sizes of the street shoes of the patients.

Patients and controls had to stand quietly on one leg for at least 3 s before a hop trial was started. They were instructed to perform single-leg hopping without any restrictions on the area of ground contact. After a tone signal, patients and subjects had to hop on one leg for 20 s. Five consecutive trials of 20 s duration were recorded for each patient. In each trial, we looked for “successful” segments of 5 consecutive hops on one leg without touching the ground with the other leg. The first of these segments was used to determine mean hopping frequency (HF) and its standard deviation (HFSD). In 21 patients, a “successful” segment of 20 s duration could be detected. Three patients had shorter “successful” segments. Nevertheless, HF and HFSD could be determined. These 3 patients were included in the HOP group (*n* = 24). In 5 patients, no “successful” segment could be detected (NO-HOP group). Because of technical problems, the trials of one patient could not be analyzed after recording. Full data analysis was performed on only 29 patients.

For each hop of the “successful” segment, it was decided whether a small initial “impact” peak was present in addition to the usual “active” peak (see Figure 1). The mean time to peak of the “active” peak (PT) and its standard deviation (PTSD), mean amplitude of the “active” peak (PA) and its standard deviation (PASD), and mean duration of ground contact (DFC) per foot and its standard deviation (DFCSD) was determined (see Figure 1). For the sake of comparison, all GRF-curves of a “successful” segment were time normalized to 100% and superimposed.

### 2.5. Statistics

A two-group ANOVA was performed to compare age, body height (BH) and body weight (BW) between the entire patient cohort and the controls. Another two-group ANOVA was performed to compare hopping parameters (HF, HFSD, DFC, DFCSD, PA, PASD, PA/BW, PA/BWSD, PT, PTSD) between the HOP group and the controls. A further two-group ANOVA compared age, body height (BH) and body weight (BW) between the HOP group and the controls. Chi-squared testing was used to compare sex distribution between the entire patient cohort and the controls, to compare patients’ distribution in the HOP- and NO-HOP subgroup, to compare the number of patients with a positive N-MotS in the HOP- and NO-HOP subgroup and to compare the number of patients with neuropsychiatric symptoms in the HOP- and the NO-HOP subgroup. The Kendell-tau test was used for non-parametric comparison of TSC, MotS and N-Mots between the HOP- and the NO-HOP subgroups. Correlations were also performed non-parametrically using Spearman’s Rho. All tests were part of the commercially available statistics package SPSS (version 25: IBM Analytics, Armonk, NY, USA).

## 3. Results

### 3.1. Comparison of the WD-Patients and the Control Subjects

Demographical data of the patient cohort and the control group are compared in Table 1, part A. Patients were perfectly sex-matched by the controls. There was no significant difference in age, body height and body weight. This is important for the frequency of hopping (HF) which is directly related to body mass. Five patients were unable to hop. Elimination of the five patients who were unable to hop (see below) from the WD patient cohort did not significantly alter mean age, mean body height and mean body weight.

### 3.2. Treatment-Related Data and Clinical Examination of the WD-Patients

Age at diagnosis ranged from 11 to 36 years (mean: 22.0, SD: 6.8 yrs). Eleven patients were females and nineteen were males, fifteen patients were treated with D-penicillamine (DPA: 600–1800 mg), fourteen with trientine (TRIEN: 600–2100 mg), five with 150–300 mg zinc in addition to DPA and TRIEN and one patient with 300 mg zinc only. Treatment duration varied between 31 and 376 months.

Bradykinesia of tongue or hand or finger movements was present in 27 (90%) patients, mild dysarthria in 20 (66%), tremor in 14 (47%), mild to moderate dystonia of upper or lower extremity in 12 (38%), dysmetric or ataxic movements of the upper or lower extremity in 12 (38%), gait abnormalities in 6 (20%) and oculomotor disturbances in 3 (10%) patients. Among the non-motor symptoms, reflex abnormalities were observed in 10 (33%), sensory disturbances in 6 (20%) and neuropsychiatric abnormalities in 9 (30%) patients. Two of the six patients with an abnormal gait suffered from a parkinsonian gait disorder and were severely affected (TSC = 12 and TSC = 7), one patient suffered from a severe ataxic gait and was the most severely affected patient (TSC = 16), and three patients suffered from a mild dystonic gait (TSC = 7, TSC = 7, TSC = 4). Five of these six patients were unable to hop (see below). Only the patient with a mild dystonic gait disorder (TSC = 4) was able to hop.

### 3.3. Ability to Perform Single-Leg Hopping

In Figure 2, attempts to perform single-leg hopping on the left (upper two traces) and on the right leg (lower two traces) are demonstrated for a healthy subject (left side), a moderately affected WD patient (middle parts) and a severely affected WD patient (right side). The corresponding time normalized, superimposed GRF-curves are shown in Figure 3. In the healthy subject, GRF-curves were highly reproducible and had one broad peak, the so-called “active peak” (left sides of Figure 2 and Figure 3). In the moderately affected WD patient, the GRF-curves were still reproducible but more variable (middle parts of Figure 2 and Figure 3) and had an additional sharp initial peak, the so-called “impact peak”. There was no foot ground contact of the contralateral foot during single-leg hopping in the healthy subject and the moderately affected patient (Figure 2, left and middle part). The severely affected WD patient was unable to perform five consecutive hops in place, neither with the right nor with the left leg (right sides of Figure 2 and Figure 3.

Five out of twenty-nine patients (NO-HOP-group) were unable to perform single-leg hopping, neither on the right nor on the left leg. Four of these five patients were females. These five patients were scored to have gait abnormalities during a clinical examination. One of these patients was unable to stand quietly on one leg for three seconds prior to hopping. In two patients, both feet had simultaneous ground contact after a hop, and two patients never succeeded to perform five repetitive hops with both legs in the air during the hops (as demonstrated in Figure 2 right side).

### 3.4. Comparison of the NO-HOP- and the HOP-Subgroup

When patients were split up into subgroups according to severity (MILD-, MOD-, SEV subgroup, see methods), chi-squared testing revealed a significant influence of severity on the ability to hop. The distribution of patients across severity subgroups was significantly different between the NO-HOP and the HOP subgroup (*p* < 0.002; Table 2, part A).

The percentage of patients with a positive N-Mots was significantly higher (*p* < 0.002; Table 2, part B) in the NO-HOP group. The percentage of patients with neuropsychiatric symptoms who are known to be less compliant compared to the rest of the patients [28] was significantly higher (*p* < 0.001; Table 2, part C) in the NO-HOP group. A two-group comparison by means of the Kendall-tau test revealed significant higher (*p* < 0.001) severity of symptoms (scored by TSC, MotS and N-MotS) in the NO-HOP group compared to the HOP-group (Table 2, part D).

### 3.5. Comparison of Single-Leg Hopping in Patients with WD and Controls

A two-group ANOVA did not reveal significant differences in the frequency of hopping (HF; Table 1, part B) and duration of foot contact (DFC) between the HOP and the control group. Peak amplitudes were slightly, but not significantly lower in the HOP group (PA: right leg: *p* = 0.164, n.s.; left leg: *p* = 0.198, n.s.). After the normalization of PA due to body weight (BW) (see Table 1, part B), this tendency towards lower values in the HOP group was enhanced, but the ANOVA did not show significant differences. The variability of peak amplitude (PASD: right leg: *p* = 0.153, n.s.; left leg; *p* = 0.233, n.s.) did not show significant difference. Only the times to the “active” peak were significantly lower (PT: right leg: *p* < 0.014; left leg: *p* < 0.022; Table 1, part B) in the patients compared to the controls.

Twelve patients out of the HOP group had an “impact peak” in all trials of both sides. Only six patients did not have any “impact peak” at all; their GRF-curves of hopping were completely normal.

In five patients of the HOP group with N-MotS > 0, a significant (*p* < 0.035) positive correlation between N-MotS and HF was found. PA was significantly lower (*p* < 0.048) in these five patients compared to the rest (*n* = 19) of the HOP group.

### 3.6. Correlation of Laboratory Findings and Parameters of Hopping

The parameters of hopping (PT, PA, DFC) and their standard deviations (PTSD, PASD, DFCSD) did not correlate with most of the l0 parameters derived from blood or urine analysis. Only the intraindividual variability of the amplitude of the “active” peak (PASD)was significantly negatively correlated with the copper concentration in the 24 h-urine collected under medication (r = −0.478; *p* < 0.018; Figure 4; data of the right leg are presented). The higher the urinary copper concentration and the copper excretion, the more reproducible single-leg hopping was.

## 4. Discussion

In the present study, single-leg hopping is analyzed in 29 long-term treated patients with neurological Wilson’s disease. In patients being able to hop (*n* = 24), the time to peak of GRF-curves were significantly shorter than in controls. Twelve patients had an “impact” peak in all GRF-curves. Five patients with a significantly higher disease severity were unable to hop.

### 4.1. Learning to Hop and the Age of Onset of Symptoms in WD

Single-leg hopping is learnt from age three on and is optimized during the next 10 years [17,18,19]. Because (i) neurological symptoms in WD usually manifest after the age of 17 and not before the age of 11 [32] and (ii) patients with a walking deficit during early childhood had been excluded (see Methods) and (iii) mean age at diagnosis of WD in the present cohort was 22 years, possibly all participants in the present study had developed nearly normal single-leg hopping. However, since intact copper homeostasis is important for the developing brain, especially for the cerebellum [33], subtle deficits of structure and function of the cerebellum in the WD-patients and the development of mild clinical cerebellar signs during adolescence cannot be completely excluded [34,35,36].

### 4.2. Response to Therapy

It was reported that gait abnormalities respond well to copper elimination therapy [6]. In normal brains, iron, copper and zinc show a complex distribution [37]. In WD this physiological distribution of heavy metals and glucose metabolism [5] is disturbed. High levels of copper lead to morphological and behavioral alterations of the rat cerebellum and balance problems. Therefore, it does not surprise us that gait abnormalities improve with copper elimination therapy [6,35].

Response of hopping to therapy in WD was not mentioned so far. The impact of severity of WD and the significant correlation between the variability of the peak amplitude (PASD) of the GRF-curves and copper concentration in the 24 h-urine (Figure 4) may be a hint that in WD, not only gait in general but also single-leg hopping, responds to copper elimination quite well.

The majority (24/29 > 82%) of the long-term treated WD-patients with a neurological manifestation in the present study managed to perform the complex motor task of single-leg hopping. This is a good message for all patients with WD and their treating physicians, which has relevant implications on patient management since it will motivate patients to take their medication and increase adherence to therapy and therapy monitoring.

### 4.3. High Percentage of Patients with an Initial “Impact” Peak

Five out of twenty-nine patients were unable to hop. This was already detected during clinical investigation. For the detection of the inability to hop, no quantitative measurement is necessary.

However, for the detection of whether an “impact peak” was present or not, the measurement of ground forces was necessary. The interpretation of the “impact peak” is complex. It may be interpreted as an artefact and may result from the slipping of the CDG^®^ shoes over the street shoes of the patients or from slipping of the feet of the patients in their street shoes.

However, it may also reflect reflex abnormalities. Variation of the height of a force platform and application of transcranial magnetic stimuli to the motor cortex even demonstrated that the early peak around 45 ms of the electromyographic activity of the soleus during hopping could be modified by change of the afferent feedback and the central drive [38]. Therefore, the occurrence of the “impact” peak cannot be explained by a single, obvious reason.

A closer look at the literature showed that the occurrence of the “impact” peak during running and hopping is well known and also occurs in robots that are able to run and hop. Engineers have already tried to present a biomechanical interpretation for the “impact” peak and have given different names to these peaks (first and second peak [39,40], impact and active peak [41,42,43] as used in the present paper, passive and active peak [44], and impact and propulsion peak [45], for an overview see [46]). It has been shown that while the “active” peak can be predicted by a simple one-body mass-spring model, one-body models are not capable of accurately predicting the “impact” peak [46]. Today the four-body model developed by Liu and Nigg (LN model [39,47]) is probably the most widely used multibody mass-spring-damper model of the human body during hopping and running [46]. Quantification of the “impact” peak is difficult, especially when hops are performed with a high variability as in WD.

For the present study, it is important to know that the occurrence of the “impact” peak may be a hint of difficulties in coordinating the different body segments as one unit. Therefore, the occurrence of an “impact” peak in 12 patients in the HOP group together with 5 patients in the NO-HOP group indicates a subtle or clinical manifest coordination deficit in about 17 out of 29 (=59%) long-term treated WD patients. This is within the reported range of percentages of abnormal gait (10–75%) during the initial investigation of WD patients before the onset of treatment [6,7,12] and is close to the percentage of WD patients with clinical cerebellar impairment (*n* = 12) or gait abnormalities (*n* = 6): *n* = 18 (18/29 = 62%). However, impact peaks were also found in patients without cerebellar impairment.

It would be interesting to see whether WD patients showed similar characteristics of immature spring function of their ankle and knee joints during hopping as children when compared to healthy adults and preadolescent children who are still developing the adult-like spring function [48].

### 4.4. Analysis of Hopping Frequency in WD

Hopping frequency (HF) mainly depends on the stiffness of the muscular system and the body mass. It is highly reproducible and similar across species and different hopping or jumping conditions [25,49,50,51]. Therefore, it does not surprise us that hopping frequency was nearly the same in the control subjects and those WD patients who were able to hop (Table 1, part B).

However, in a small subgroup of patients (*n* = 5) who were able to hop but had a positive non-motor score, a significant positive correlation between the N-MotS and HF was found. These patients produced lower peak amplitudes and shorter times to peak, thus significantly reducing the mean PT of the entire HOP group as presented in Table 1, part B. At least three of these five patients had enhanced reflexes. Therefore, the correlation between N-MotS and HF and the shorter times to peak reflects an increase in stiffness of the neuromuscular system, probably due to dystonia or spasticity.

### 4.5. Variability of the Peak Amplitude

The number of patients who were unable to hop was over-represented in the subgroup of patients with psychiatric abnormalities. These patients are well-known to be less compliant compared to the rest of the patients [30]. The inverse relationship between the variability of PA and the copper excretion indicates that the stabilization of hopping [52] can be improved by copper elimination therapy [53]. It will be interesting to analyze in further studies whether other motor activities affording precise motor tuning such as speaking, e.g., [54], will improve in parallel when hopping and other motor activities are analyzed in parallel.

These studies should also include high resolution MRI-scanning (see [55]).

## 5. Conclusions

Most of the long-term treated patients with a neurological manifestation in the present study were able to perform the complex motor task of single-leg hopping. This is an encouraging message for all patients with a neurological manifestation of WD who are usually severely affected when WD is diagnosed and the disease-specific treatment is initiated. Those WD patients who remain unable to hop can easily be detected during clinical investigation.

Quantitative measurement and analysis of the shapes of GRF-curves allow us to detect the so-called “impact” peak, which may indicate a subclinical coordination deficit in long-term treated WD patients. The inverse relationship between variability of time to peak, which is a measure for hopping stabilization and copper excretion, indicates the possibility to further improve the mild motor coordination deficit in WD. The present pilot study also suggests that single-leg hopping may be a useful tool to detect subclinical limb/trunk coordination deficits in other movement disorders [56].

### Strengths and Limitations of the Study

Hopping was analyzed by means of the Infotronic^®^ CDG^®^ gait analysis system, which is easy to handle but does not allow the measurement of joint angles and movements of parts of the body in space. The present study confirms the presence of a mild forefoot control deficit during free-walking [16]. However, it does not specify whether the coordination deficit in WD results from a limb control deficit or from a control deficit of trunk and head in addition. Therefore, further studies using a motion capture system in combination with a force platform are recommended to analyze the coordination deficit during hopping in patients with WD in more detail.

## Figures and Tables

**Figure 1 medicina-58-00249-f001:**
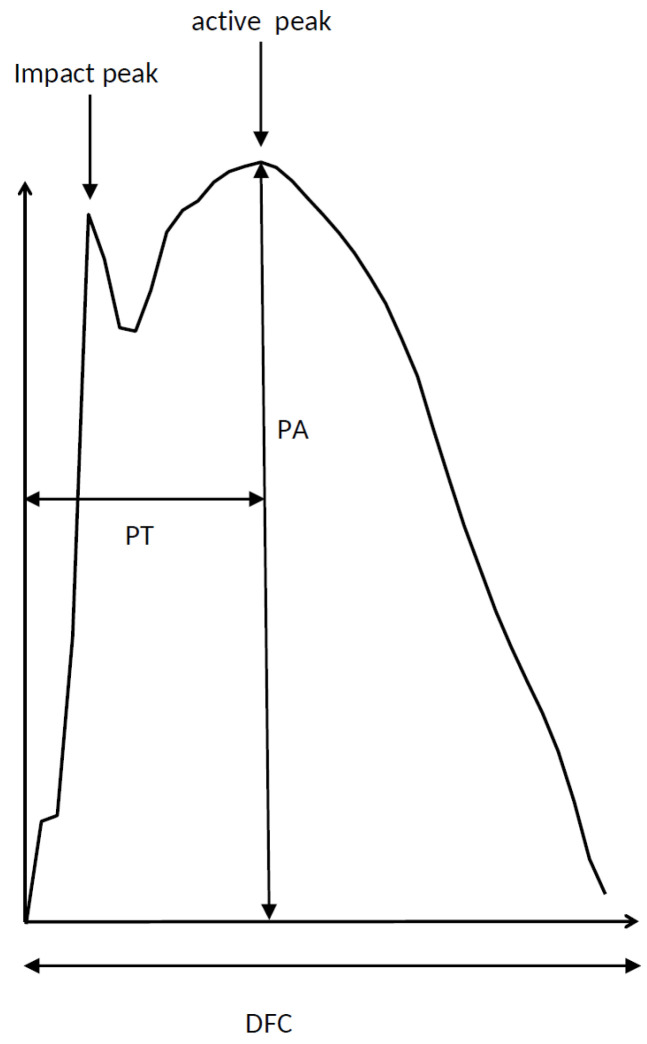
Parameters derived from ground reaction forces (GRF-curves) during single-leg hopping were: duration of foot contact (DFC), time to peak (PT) and peak amplitude (PA). For the sake of comparison, GRF-curves were normalized to 100% duration (after DFC had been determined). Especially in the GRF-curves of the moderately and severely affected patients, a second sharp peak (impact peak) was reproducibly present in addition to the usual peak (active peak). In the literature, other names are also given to these two peaks (see Section 4).

**Figure 2 medicina-58-00249-f002:**
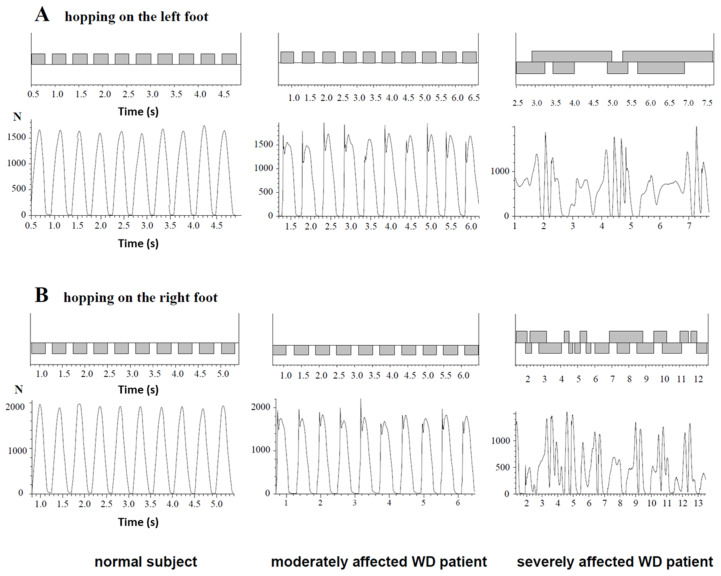
Attempts of single-leg hopping are presented for a normal subject (**left side**), a moderately affected WD patient (**middle part**) and a severely affected WD patient (**right side**). The **first trace** shows segments of foot contact to the ground during single-leg hopping on the left leg, the **second trace** reveals the corresponding GRF-curves of the left leg, The **third trace** shows segments of foot contact to the ground during single-leg hopping on the right leg, the **fourth trace** reveals the corresponding GRF-curves of the right leg. The normal subject hops with only one “active peak”, the moderately affected patient produces GRF-curves with a first sharp “impact” peak and a second broad “active” peak. The severely affected patient is not able to hop on one leg, neither on the right nor on the left leg.

**Figure 3 medicina-58-00249-f003:**
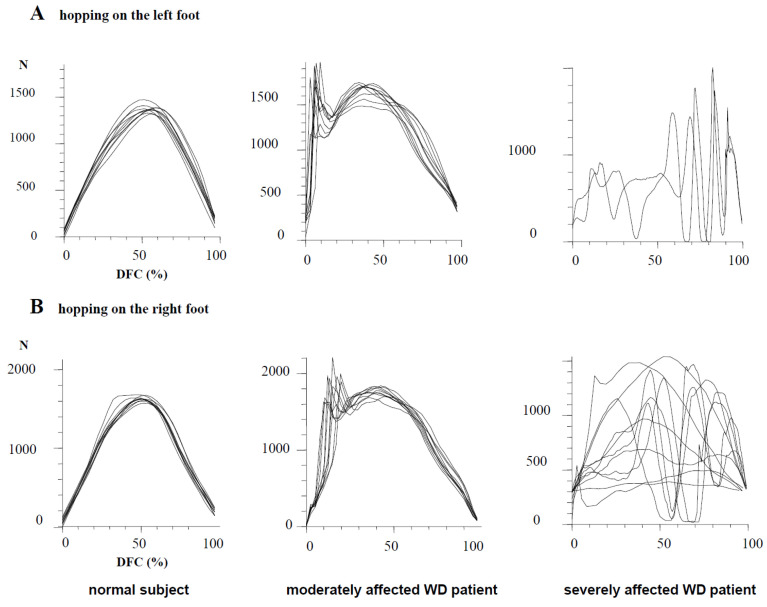
Time normalized GRF-curves (**upper trace**: curves of the left leg, **lower trace**: curves of the right leg) of the same normal subject and the same patients as in Figure 2. Superposition of the time normalized GRF-curves demonstrates the reproducibility of the GRF-curves in the normal subject and the moderately affected patient with only one broad “active” peak in the normal subject (**left side**) and an additional sharp initial “impact” peak in the moderately affected patient (**middle part**). In the severely affected patient (**right side**), no reproducible curve and peak can be detected.

**Figure 4 medicina-58-00249-f004:**
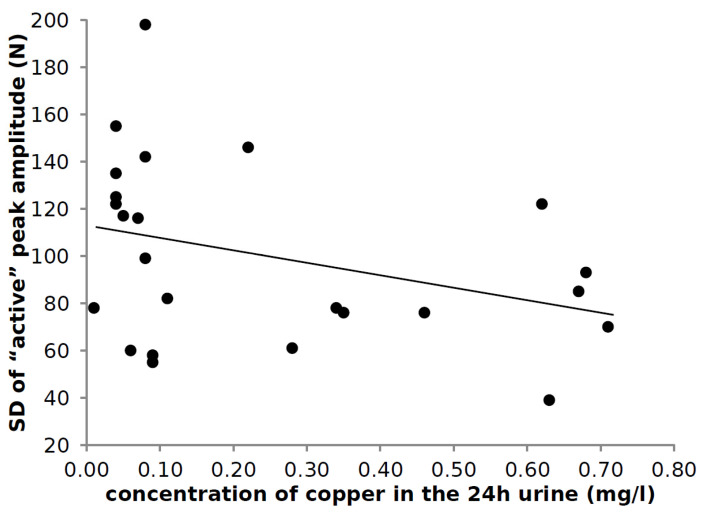
Correlation between the mean variability per patient of the peak amplitude and the copper concentration in the 24 h-urine (collected under medication). The higher the urinary copper concentration and the copper excretion, the smaller is PASD and the more reproducible the hopping on one leg was.

**Table 1 medicina-58-00249-t001:** Demographical and hopping data of WD patients and controls.

**Part A**					
**Parameter**			**WD-Patients** ***n* = 30**	**Control Subjects** ***n* = 30**	***p*-Value**
Age (years)		mean (SD)	34.2 (11.0)	32.8 (8.3)	*p* = 0.571; n.s.
		range	15–56	14–51	
Sex		m/f	19/11	19/11	*p* = 1.000; n.s.
Body height (cm) BH		mean (SD)	177.8 (10.3)	176.9 (9.5)	*p* = 0.730; n.s.
		range	156–196	160–194	
Body weight (kg) BW		mean (SD)	72.7 (1.5)	73.6 (14.8)	*p* = 0.830; n.s.
		range	48–126	51–105	
**Part B**					
**Parameter**			**HOP-Group** ***n* = 24**	**Control Subjects** ***n* = 30**	***p*-Value**
HF (1/s)	L	mean (SD)	1.94 (0.29)	1.95 (0.18)	*p* = 0.681; n.s.
	R	mean (SD)	1.90 (0.33)	1.92 (0.18)	*p* = 0.841; n.s.
DFC (ms)	L	mean (SD)	373 (94)	368 (50)	*p* = 0.823; n.s.
	R	mean (SD)	362 (90)	357 (46)	*p* = 0.675; n.s.
PA/BW (N/kg)	L	mean (SD)	23.25 (4.95)	24.85 (3.20)	*p* = 0.159; n.s.
	R	mean (SD)	23.12 (4.82)	25.04 (3.02)	*p* = 0.080; n.s.
PT (ms)	L	mean (SD)	127.8 (27.2)	144.4 (24.6)	*p* < 0.022
	R	mean (SD)	126.9 (30.2)	146.8 (27.5)	*p* < 0.014

mean = mean value; SD = standard deviation; m = male; f = female; BH = body height; BW = body weight; L = left leg; R = right leg; HF = frequency of hopping; DFC = duration of foot contact; PA/BW = amplitude of the “active” peak divided by body weight; PT = time to “active” peak; n.s. = not significant.

**Table 2 medicina-58-00249-t002:** Comparison of 24 patients being able to hop and 5 patients being unable to hop.

**Part A**				
**Parameter**		**NO-HOP Subgroup (*n* = 5)**	**HOP Subgroup (*n* = 24)**	**Level of Significance**
MIL-group	number of pats.	0	10	
MOD-group	number of pats.	1	10	
SEV-group	number of pats.	4	4	*p* < 0.002
**Part B**				
Parameter				
N-MotS = 0	number of pats.	0	19	
N-MotS > 0	number of pats.	5	5	*p* < 0.002
**Part C**				
No psychiatric symptoms	number of pats.	0	21	
With psychiatric symptoms	number of pats.	5	3	*p* < 0.001
**Part D**				
TSC	MV/SD	10.8/3.6	3.4/2.4	*p* < 0.0001
MotS	MV/SD	7.4/3.0	3.0/2.0	*p* < 0.0001
N-MotS	MV/SD	3.4/0.9	0.4/0.9	*p* < 0.0001

NO-HOP = patients being unable to hop; HOP = patients being able to hop; MIL = mildly affected patients; MOD = moderately affected patients; SEV = severely affected patients (see Methods); TSC = total score; MotS = motor score; N-MotS = non-motor score (see Methods); MV = mean value; SD = standard deviation.

## Data Availability

Data available on request due to restrictions (e.g., privacy or ethical). The data presented in this study are available on request from the corresponding author.

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
