# Peer review of "Analysis of Single-Leg Hopping in Long-Term Treated Patients with Neurological Wilson’s Disease: A Controlled Pilot Study"

_medicina, 2022, doi:10.3390/medicina58020249_

Round 1

Reviewer 1 Report

discussion needs improvement with more literature review as per r the findings in the manuscript

Author Response

Reviewer 1 is right. We have rewritten the discussion, have added more literature on hopping and the cerebellum as suggested by reviewer 3. 

Reviewer 2 Report

There are several papers (lats 5 years) describing neurological involvement of WD, including gait disturbances, posture , and others. . I'm not sure if hopping on one leg test is adequate for WD and provide new data for WD patients.

The group of WD patients is small, there are no data about neurological severity of tases patients(neurological form: dystonia? parkinsonism, etc), as well no about severity scored in world acceptable scales GAS, UWDRS ( accepted by MDS),. There are no data about brain MRI (brain injury in WD), 

The idea of study isn not clearly presented, the usefulness of used test not presented in the disorders, conclusions are also limited - what is practical point from this study?

Author Response

There are several papers (lats 5 years) describing neurological involvement of WD, including gait disturbances, posture, and others.. I'm not sure if hopping on one leg test is adequate for WD and provide new data for WD patients.

The group of WD patients is small, there are no data about neurological severity of tases patients(neurological form: dystonia? parkinsonism, etc), as well no about severity scored in world acceptable scales GAS, UWDRS ( accepted by MDS),. There are no data about brain MRI (brain injury in WD), 

The idea of study isn not clearly presented, the usefulness of used test not presented in the disorders, conclusions are also limited - what is practical point from this study?

As suggested by reviewer 2 we have added more recent literature on neurological impairment in WD.

We have explained in detail why single-leg hopping is an useful tool to analyse subtle motor coordination impairment in WD.

Whether a cohort of WD-patients is estimated to be small or large depends on the test being analysed. This is a pilot study. Hopping has not been analysed in WD before.

Reviewer 2 is absolutely right. The results of clinical examination of the WD-patients was completely omitted. This is improved now. We are very thankful for this hint.

The idea, to detect subclinical difficulties in motor coordination in long-term treated, mildly affected WD-patients is emphasized more clearly.

Reviewer 3 Report

The present study examines the central motor system. The complexity of the force-time coordination is measured. The impairment of the cerebellar function in particular can be demonstrated using the example of copper toxicity. This should be emphasized in the discussion. In the case of the quotations on motor skills, older and own sources are mainly used. Some more recent studies should also be included here. Overall, the measurement of hopping is very interesting. I recommend publishing the results.

Pathophysiologically, the function of the cerebellum should be discussed.

Author Response

The present study examines the central motor system. The complexity of the force-time coordination is measured. The impairment of the cerebellar function in particular can be demonstrated using the example of copper toxicity. This should be emphasized in the discussion. In the case of the quotations on motor skills, older and own sources are mainly used. Some more recent studies should also be included here. Overall, the measurement of hopping is very interesting. I recommend publishing the results.

Pathophysiologically, the function of the cerebellum should be discussed.

Reviewer emphasizes the role of the cerebellum for single-leg hopping.

We therefore have modified and rewritten large parts of the discussion.

We are thankful to reviewer 3 to have raised this relevant aspect and the possibility to include this aspect in the discussion. 

Round 2

Reviewer 2 Report

I have no more comments, authors corrected the article